# Inhibitory Neural Network’s Impairments at Hippocampal CA1 LTP in an Aged Transgenic Mouse Model of Alzheimer’s Disease

**DOI:** 10.3390/ijms22020698

**Published:** 2021-01-12

**Authors:** Hyeon Jeong Seo, Jung Eun Park, Seong-Min Choi, Taekyoung Kim, Soo Hyun Cho, Kyung-Hwa Lee, Woo Keun Song, Juhyun Song, Han-Seong Jeong, Dong Hyun Kim, Byeong C. Kim

**Affiliations:** 1Department of Biomedical Sciences, Graduate School, Chonnam National University, Gwangju 61186, Korea; eowk1689@nate.com; 2Department of Biomedical Science, College of Natural Sciences, Chosun University, Gwangju 61452, Korea; jepark@chosun.ac.kr; 3Department of Integrative Biological Sciences & BK21 FOUR Educational Research Group for Age-Associated Disorder Control Technology, Chosun University, Gwangju 61452, Korea; 4Department of Neurology, Chonnam National University Medical School, Gwangju 61469, Korea; drchoism@gmail.com (S.-M.C.); k906141h@hanmail.net (S.H.C.); 5Department of Neurology, Chonnam National University Hospital, Gwangju 61469, Korea; grasshaha@naver.com; 6Department of Pathology, Chonnam National University Medical School & Hwasun Hospital, Hwasun 58128, Korea; mdkaylee@jnu.ac.kr; 7Cell Logistics and Silver Health Research Center, School of Life Sciences, Gwangju Institute of Science and Technology, Gwangju 61005, Korea; wksong@gist.ac.kr; 8Department of Anatomy, Chonnam National University Medical School, Hwasun 58128, Korea; juhyunsong@jnu.ac.kr; 9Department of Physiology, Chonnam National University Medical School, Hwasun 58128, Korea; jhsjeong@hanmail.net; 10Department of Health Sciences, The Graduate School of Dong-A University, Busan 49236, Korea

**Keywords:** Alzheimer’s disease, hippocampus, CA1, hippocampal long-term potentiation, NRG1-ErbB4 signaling

## Abstract

Alzheimer’s disease (AD) is a neurodegenerative disorder characterized by a rapid accumulation of amyloid β (Aβ) protein in the hippocampus, which impairs synaptic structures and neuronal signal transmission, induces neuronal loss, and diminishes memory and cognitive functions. The present study investigated the impact of neuregulin 1 (NRG1)-ErbB4 signaling on the impairment of neural networks underlying hippocampal long-term potentiation (LTP) in 5xFAD mice, a model of AD with greater symptom severity than that of TG2576 mice. Specifically, we observed parvalbumin (PV)-containing hippocampal interneurons, the effect of NRG1 on hippocampal LTP, and the functioning of learning and memory. We found a significant decrease in the number of PV interneurons in 11-month-old 5xFAD mice. Moreover, synaptic transmission in the 5xFAD mice decreased at 6 months of age. The 11-month-old transgenic AD mice showed fewer inhibitory PV neurons and impaired NRG1-ErbB4 signaling than did wild-type mice, indicating that the former exhibit the impairment of neuronal networks underlying LTP in the hippocampal Schaffer-collateral pathway. In conclusion, this study confirmed the impaired LTP in 5xFAD mice and its association with aberrant NRG1-ErbB signaling in the neuronal network.

## 1. Introduction

Alzheimer’s disease (AD), which accounts for more than 60% of dementia cases, is characterized by the pathological deposition of amyloid-β (Aβ) peptides in the hippocampus [1,2,3]. The accumulation of these deposits induces hippocampal atrophy [4] and impairs memory and cognitive functions [5]. Although the pathological characteristics of AD are known to be amyloid plaques and neurofibrillary tangles [6], the cause of AD-induced dementia remains undetermined. AD research has primarily focused on amyloid plaques, regardless of their etiology. Specifically, previous studies have implicated the following three mutations in the excessive accumulation of Aβ peptides as amyloid plaques and in the onset of familial AD [7]: amyloid precursor protein (*APP*), presenilin-1 (*PSEN1*), and presenilin-2 (*PSEN2*).

Normal aging is associated with dementia-related pathologies and cognitive decline similar to those observed in AD of aged individuals. Although both feature synaptic loss, it is possible to distinguish between decreased memory in normal aging and AD based on the loss of parvalbumin (PV) neurons [8]. Indeed, a decrease in the number of PV neurons has only been observed in AD models [8,9]. Excessive accumulation of toxic substances causes a dramatic decrease in the number of neurons in the brain, leading to a reduced number of PV interneurons in the hippocampus of aged AD mice [9,10], as well as age-related decrease of hippocampal PV neurons in human AD and animal models of AD [11,12]. PV interneurons are abundant among gamma amino butyric acid (GABA) as their output are called GABAergic neurons, and inhibitory neurons that provide feedback and feed forward inhibition to excitatory pyramidal neurons in multiple brain regions, including the hippocampus [13,14,15]. Loss of PV interneuron’s function is associated with decreased GABAergic transmission and not only reduces inhibitory control over pyramidal cell activity, but also reduces coordinated activities between brain networks [16].

The present study investigated hippocampal long-term potentiation (LTP) mechanisms to examine the effects of inhibitory networks on brain activity. GABAergic transmission has been reported to modulate hippocampal synaptic plasticity [17,18,19] and plays an important role in neuregulin 1 (NRG1)-dependent LTP of hippocampal cornu ammonis (CA3–CA1) synapses. Endogenous NRG1, which mediates the inhibitory effects, is known to bind to and act on the receptor tyrosine-protein kinase ErbB4. NRG1 regulates neuronal excitatory synaptic activity and synaptic plasticity in the adult brain and has been implicated in neural development including circuitry generation and axon ensheathment [20]. The involvement of NRG1 in LTP has been demonstrated by the ablation of *ErbB4* gene in PV-expressing GABAergic interneurons of mice [21]. Moreover, the treatment of hippocampal slices with NRG1 rapidly inhibits LTP expression in Schaffer collateral-CA1 synapses [22]. However, this inhibition by NRG1 did not occur in animals with an *ErbB4* gene deletion [21,23]. Neutralization of both endogenous NRG1 and ErbB4 mutations enhances hippocampal LTP [21,23] and is regulated by NRG1 in completely unresolved reverse LTP [24]. This observation suggests that the NRG1-ErbB signaling is also involved in hippocampal CA3-CA1 synaptic plasticity.

This study investigated the role of NRG1-ErbB signaling in the mechanisms underlying Aβ-oligomer-mediated synaptic function deficiencies in 5xFAD mice, which feature *APP* and *PSEN1* transgenics to model the features of AD. The discovery of similarly impaired neural networks in the presently used 5xFAD mice and the previously investigated TG2576 mouse model of AD indicates that amyloid pathology significantly influences NRG1-ErbB signaling.

## 2. Results

### 2.1. PV Interneurons Decreased in Hippocampus CA1-3 Region of Aged 5xFAD Mice

PV expressing GABAergic inhibitory neurons are essential for hippocampal neuronal activity [13,14,15], and aged AD mice have a reduced number of PV interneurons in the hippocampus relative to wild-type (WT) mice [9,10]. Therefore, we evaluated the number of PV interneurons in the hippocampal CA1-CA3 region of adult (6-month-old) and aged (11-month-old) AD mice (Figure 1A). To determine the percentage of PV interneurons, neuronal nuclei (NeuN) and PV interneurons were counted in the entire CA subfield (Figure 1), and the ratio of PV interneurons to NeuN cells was calculated. The number of PV-positive neurons was lower in aged 5xFAD mice than in adult 5xFAD mice and WT mice (Figure 1B,C). The ratio of PV interneurons to NeuN-positive cells was significantly reduced in aged mice relative to adult mice (Figure 1F). Importantly, the ratio was not significantly different between adult and aged WT littermates, but significantly decreased in 5xFAD mice, relative to WT mice (Figure 1F). Therefore, these reductions in PV neurons are not attributable to aging alone, but likely involve amyloid pathology.

### 2.2. Evaluation of Synaptic Plasticity in Aged AD Mice

Approximately 26% of the GABAergic neurons in the hippocampal CA1 region are PV interneurons that provide inhibitory input to hippocampal pyramidal neurons [25,26,27]. Because of the reduced inhibitory input caused by PV loss in aged AD mice, LTP-induced field excitatory postsynaptic potential (fEPSP) is not inhibited and may be maintained with aged in 5xFAD mice. We thus compared LTP in the hippocampal CA1 area between the three different age groups. The 5xFAD mouse is a well-established AD model with high Aβ levels and severe memory impairments. This transgenic animal starts accumulating soluble Αβ in the brain at about 1.5 months of age, and LTP is abolished at 6 months [7,28]. In 3-month-old animals, LTP was observed in both WT and 5xFAD mice, and the increased fEPSPs levels were not significantly different (TG: 167.96 ± 12.79%, *n* = 5, closed circle; WT: 170.48 ± 8.53%, *n* = 4, open circle, *p* > 0.05; Figure 2A,D). The increased fEPSPs induced by high-frequency stimulation (HFS) of the Schaffer collateral pathway were maintained for 2 h in adult WT littermates. 5xFAD mice returned to baseline in 6-month-old 5xFAD mice (TG: 102.09 ± 3.97%, *n* = 5, closed circle; WT: 162.44 ± 6.50, *n* = 5, open circle, *p* < 0.0001; Figure 2B,E). The results obtained with 6-month-old mice match those of a previous study, which reported LTP in 5xFAD mice [28]. By contrast, increased fEPSP was maintained for up to 2 h after HFS in aged (11-month-old) 5xFAD mice, and there was no significant difference in the level of fEPSP enhancement between aged 5xFAD and their WT littermates (TG: 179.29 ± 17.97%, *n* = 5, closed circle; WT: 168.57 ± 8.90%, *n* = 5, open circle, *p* > 0.05; Figure 2C,F). These data show that the maintenance of LTP-induced fEPSP in aged 5xFAD may be associated with the loss of PV interneurons in the hippocampus. Additionally, our findings are similar to the whole-cell patch-clamp recording results published in previous studies, showing that the frequency and amplitude of spontaneous excitatory postsynaptic potentials (sEPSPs) in the hippocampal stratum radiatum were similar between 11~15-month-old transgenic and wild-type mice [29] (below the corrected Figure 2).

### 2.3. Effect of NRG1-ErbB Signaling in Aged AD Mice

To further examine how the NRG1-ErbB signaling pathway regulates synaptic plasticity in aged 5xFAD mice, we perfused hippocampal slices with recombinant NRG1 peptide and observed the expression of hippocampal LTP by activating ErbB4 receptor kinase in the PV interneurons [21,30,31]. NRG1-induced LTP inhibition was assessed by measuring fEPSP slopes in the hippocampal slices after HFS stimulation. LTP was induced at 3 months of age in both WT and 5xFAD mice (Figure 2A). Perfusion with NRG1 peptide (1 nM) 20 min before HFS significantly reduced LTP in 3-month-old 5xFAD mice and their WT littermates as compared with those perfused with phosphate-buffered saline (PBS; control) (TG PBS: 151.09 ± 14.55%, *n* = 4, dark gray circle; WT PBS: 150.00 ± 3.53%, *n* = 5, light gray circle; TG NRG1: 114.20 ± 10.07%, *n* = 5, black circle; *p* < 0.0001 compared with TG PBS; WT NRG1: 109.70 ± 9.92%, *n* = 5, white circle, *p* < 0.05 compared with WT PBS; Figure 3A,D). Perfusion with NRG1 in both 3-month-old 5xFAD and their WT littermates induced a reduction of LTP and showed expression of NRG1-ErbB4 signaling.

NRG1-induced LTP reduction was also tested in 11-month-old (aged) 5xFAD mice and their WT littermates. We confirmed that NRG1 perfusion inhibited LTP expression in aged WT. However, this was not observed in aged 5xFAD mice (TG PBS: 178.17 ± 12.60%, *n* = 5, dark gray circle; WT PBS: 167.10 ± 24.28%, *n* = 5, light gray circle; TG NRG1: 188.22 ± 16.31%, *n* = 5, black circle, *p* > 0.05, as compared with TG PBS; WT NRG1: 132.64 ± 14.84%, *n* = 5, white circle, *p* < 0.05, as compared with WT PBS; Figure 3C,F). Our results indicate that the NRG1-expressed was not LTP-expressed fEPSP in aged 5xFAD mice, and decreased PV interneurons. NRG1-ErbB signal impairment was only present in aged 5xFAD mice. As NRG1-ErbB signaling should be maintained in adult AD animal models, we investigated the inhibitory effect of ErbB4 receptor activity on hippocampal synaptic plasticity. LTP was induced in PD158780 as an inhibitor in ErbB receptor kinase-treated slices. We found that fEPSPs were maintained for more than 2 h (TG PBS: 118.51 ± 12.70%, *n* = 5, dark gray circle; WT PBS: 157.24 ± 2.15%, *n* = 5, light gray circle; TG PD157870: 191.06 ± 10.25%, *n* = 5, black circle, *p* < 0.0001, as compared with TG PBS; WT PD158780: 159.36 ± 8.01%, *n* = 4, white circle, *p* < 0.0001, as compared with WT PBS; Figure 3B,E). ErbB inhibition improved the level of LTP and compromised NRG1-ErbB signaling in adult animals (Figure 3B) (below the corrected Figure 3).

Interfering with endogenous NRG1-ErbB signaling by PD158780 treatment is already known to enhance fEPSP levels at the hippocampal synapses [24]. Thus, we tested whether PD158780 perfusion after LTP expression could rescue the LTP impairment in adult 5xFAD mice. Perfusion with 10 µM of PD158780 for 1 h after LTP expression reversed the decrease in LTP; moreover, enhanced fEPSPs were maintained for an additional 2 h (TG PD158780: 158.85 ± 14.22%, *n* = 6, closed circle; Appendix A). These results suggest that ErbB signaling, which is responsible for inhibitory activity, was maintained in adult AD mice, and the inhibited LTP reappeared after blocking the ErbB receptor.

### 2.4. Evaluation of Depontentiation-Induced NRG1-ErbB Signaling in Aged 5xFAD Mice

NRG1-ErbB signaling may participate in other forms of synaptic plasticity [32,33]. Another stimulation protocol leads to depotentiation, a mechanism for preserving synaptic homeostasis at the hippocampal CA3-CA1 synapse. Depotentiation can be elicited by some forms of synaptic activity within minutes of LTP induction [34,35,36], and previous studies have demonstrated that it is related to NRG1-ErbB signaling [32,33]. To examine the effects of depotentiation in 3-, 6-, and 11-month-old 5xFAD mice, theta-pulse stimulation (TPS) was applied to the Schaffer collateral pathway in the hippocampus 5 min after HFS. HFS-induced fEPSP enhancement decreased, returning to approximately baseline after 2 h of application of TPS in both young (3-month-old) 5xFAD and their WT littermates, indicating synaptic depotentiation (TG: 105.67 ± 5.31%, *n* = 5, black circle; WT: 117.20 ± 6.76%, *n* = 4, white circle, *p* > 0.05; Figure 4A).

Furthermore, TPS-induced depotentiation was measured immediately after HFS application. The results of this experiment demonstrated LTP depotentiation with TPS (TG: 122.82 ± 11.85%, *n* = 5, black circle; WT: 122.10 ± 5.62%, white circle, *n* = 5, *p* > 0.05; Figure 4B). By contrast, marked depotentiation was not observed in ErbB receptor inhibited mice (TG PD158780: 175.02 ± 11.16%, *n* = 5, closed circle; Appendix A). These data indicate the retention of NRG1-ErbB signaling in adult 5xFAD mice, and that NRG1-ErbB activation is an inhibitory signal.

Moreover, HFS-induced LTP showed LTP depotentiation with TPS in aged WT littermate mice. However, this was maintained at elevated levels for 2 h after application of TPS inaged 5xFAD mice (TG: 201.76 ± 19.89%, *n* = 6, closed circle; WT: 136.34 ± 6.05%, *n* = 6, open circle, *p* < 0.0001; Figure 4C). The absence of depotentiation in aged 5xFAD hippocampal slices is consistent with impaired NRG1-ErbB signaling in aged 5xFAD mice. This observation suggests that synaptic functions in aged 5xFAD mice are abnormal, despite the presence of HFS-induced LTP in these animals. Considered collectively, our results indicate that activation of the NRG1-ErbB signaling pathway with NRG1 is decreased following hippocampal LTP in aged 5xFAD mice (below the corrected Figure 4).

### 2.5. Imbalance between NRG1 and ErbB4 Receptor Proteins in Aged 5xFAD Mice

A previous study showed that the overexpression of type I or type III NRG1 improves cognitive deficiencies and neuropathology in murine models of AD, suggesting that NRG1 signaling may have a significant effect on AD [37]. NRG1 expression decreases in the brains of older individuals with AD [38]. The expression levels of NRG1 and ErbB4 proteins in the hippocampus were measured in both 5xFAD and WT mice across age with Western blotting (Figure 5A). We found an age-dependent decrease in the expression level of NRG1, whereas the expression of ErbB4 receptor increased (Figure 5A). In addition, the expression levels of NRG1 and ErbB4 proteins did not differ between 3-month-old WT and 5xFAD mice; however, it became significantly imbalanced at 6 and 11 months of age (Figure 5B,C). These results suggest that imbalanced NRG1-ErbB expression affects synaptic plasticity, and is associated with amyloid pathology in AD.

### 2.6. Evaluation of Memory Impairment in Aged 5xFAD Mice

To assess memory in relation to the re-emerged LTP in 11-month-old 5xFAD mice, we performed novel object recognition tests. The preference index indicated an impairment of object recognition memory in both 6- and 11-month-old 5xFAD mice. This result demonstrates that 6-month-old 5xFAD mice have impaired memory (Figure 6B), which was consistent with the observed reduction in LTP at 6 months. However, unlike reappearing LTP observed in 11-month-old 5xFAD mice, memory impairment persisted and affected the performance of mice in behavioral experiments (Figure 6C).

## 3. Discussion

We used mutant APP and PSEN1 expressing 5xFAD mice to investigate the pathology of Aβ-induced neuronal dysfunction in AD at three different time points, at 3, 6, and 11 months [7,28,39]. 5xFAD mice represent a superior model to other AD mice models for investigating neuronal loss associated Aβ accumulation [40]. In addition, in the 5xFAD mice used in this study, a model of AD with greater symptom severity than that of TG2576 mice, PV loss was induced at a young age [30]. Loss of PV neurons in 5xFAD mice was observed at 11 months of age (Figure 1F). Importantly, the ratio of PV interneurons to NeuN cells did not differ significantly between adult (6-month-old) and aged (11-month-old) WT littermates (Figure 1F). Therefore, the reduction in PV neurons was not attributable to aging alone; moreover, this loss of PV neurons affects synaptic plasticity (Figure 2C). These results indicate that a decrease in PV neurons is associated with synaptic dysfunction in aged (11-month-old) AD mice.

PV-expressing GABAergic inhibitory neurons are essential for hippocampal neuronal activity [13,14,15]. GABAergic transmission plays an important role in NRG1-dependent LTP in the hippocampal CA3–CA1 synapses. Endogenous NRG1, which mediates inhibitory effects, is known to bind to and act on the receptor tyrosine-protein kinase, ErbB4, and regulate neuronal synaptic activity [20]. Perfusion of NRG1 to the hippocampal slices prevents LTP expression at the CA1 synapses (Figure 2A), implicating the role of NRG1-ErbB4 signaling in the modulation of synaptic plasticity in 3-month-old AD mice. However, the NRG1-ErbB4 signaling pathway did not seem to regulate synaptic plasticity in 11-month-old AD mice (Figure 2C). Inhibition of ErbB4 enhanced LTP in 6-month-old 5xFAD mice (Figure 3B). Previous studies have shown improved memory in the absence of or in the inhibition of ErbB4 in the mouse brain [41,42,43]. Absence of NRG1-ErbB in 11-month-old AD mice was associated with impaired cognitive ability despite enhanced LTP. Previous studies have shown that PV-ErbB4−/− mice have enhanced LTP in the amygdala, but experience memory impairment [44]. It remains controversial whether enhanced LTP, achieved by inhibiting ErbB, is associated with cognitive enhancement.

Unlike long-term depression (LTD), depotentiation is the response reduction affecting the responses previously increased by LTP [45]. In addition, depotentiation mechanisms are forgetting mechanisms, necessary for proper learning of new cues without catastrophically forgetting previously learned ones [46]. After LTP induction, TPS can induce potentiation within a few minutes [34,35,36]. However, the mechanisms underlying depotentiation, induced by additional stimulations after LTP induction, remain unclear. Previous experiments have shown that NRG1-ErbB signaling may participate in other forms of synaptic plasticity [32,33]. Depotentiation expression in 5xFAD mice at 3 and 6 months of age revealed the involvement of NRG1-ErbB signaling (Figure 4A,B). Furthermore, interference with NRG1-ErbB signaling did not express depotentiation (Appendix A). However, the expression of depotentiation showed the disruption of NRG1-ErbB signaling in 11-month-old 5xFAD mice (Figure 4C), indicating that NRG1-ErbB participates in depotentiation expression, which is not induced in aged 5xFAD mice, which has compromised NRG1-ErbB signaling.

Amyloid deposition in AD affects NRG1 and ErbB signaling [47], and reduced levels of NRG1 have been observed in the hippocampi of patients with AD [38]. In addition, ErbB4 immunoreactivity is reportedly increased in the brains of patients with AD [48]. In the cerebral cortex and hippocampus of APP/PSEN1 double transgenic mice, ErbB4 immunoreactivity was found to be significantly increased relative to age-matched WT controls [48]. In the hippocampi of older (11-month-old) AD mice, the levels of NRG1 and ErbB are decreased and increased, respectively (Figure 5A).

In conclusion, the decrease of PV neurons interferes with NRG1-ErbB signaling in aged AD mice. Moreover, the imbalance between NRG1 and ErbB proteins causes abnormal synaptic plasticity, and consequently affects memory function (Figure 7). ErbB4 plays a critical role in the plasticity of the excitatory synaptic function [49]. Therefore, the importance of suppressing ErbB4 is clear. However, further study is needed to identify its association with Aβ accumulation in AD-induced processes.

## 4. Materials and Methods

### 4.1. Reagents

Recombinant human NRG1-beta 1/HRG1-beta 1 EGF domain proteins (396-HB-050; R&D Systems, Minneapolis, MN, USA), reconstituted in sterile PBS containing at least 0.1% human or bovine serum albumin to a final concentration of at 100 μg/mL. PD158780 (513035; Calbiochem, Darmstadt, Germany) and reconstituted in dimethyl sulfoxide (5 mg/mL). Drugs were prepared as stock solutions, stored below −20 °C, and dissolved in artificial cerebrospinal fluid (aCSF) at least 30 min before use.

### 4.2. Animals

All procedures performed in the present study were based on the protocols published by the Institutional Animal Care and Use Committee of Chonnam National University (Project title: The study of LTP in a transgenic AD mouse model, approval number CNUHIACUC-18012, 27 September 2019). 5xFAD mice carry human *APP* and *PSEN1* transgenes with a total of five AD-linked mutations. The *APP* contains the Swedish (K670N/M671L), Florida (I716V), and London (V717I) mutations, and the *PSEN1* contains the M146L and L286V mutations. The male 5xFAD mice (034840; The Jackson Laboratory, Bar Harbor, ME, USA) were crossed with female B6/SJL F1 (B6SJL-F; Taconic Biosciences, Rensselaer, New York, USA). Mice were housed in individually ventilated cages with a 12 h/12 h light/dark cycle and a room temperature of 22–30 °C, and the animals were supplied with water and fed ad libitum.

### 4.3. Hippocampal Slice Preparation

Mice were sacrificed by cervical dislocation to minimize their suffering. The brains were carefully removed from the skull, and placed in ice-cold (4 °C) aCSF solution carbogenated with 95% O_2_ and 5% CO_2_. The solute concentrations of the aCSF (in mM) were as follows: 124 mM NaCl, 3 mM KCl, 26 mM NaHCO_3_, 1.25 mM NaH_2_PO_4_, 2 mM CaCl_2_, 1 mM MgSO_4_, and 10 mM D-Glucose. Hippocampal slices (up to 400 μm thick) were prepared using a McIlwain Tissue Chopper (Mickle Laboratory Engineering Co. Ltd., Guildford, UK). Retrieved slices were allowed to recover for 1.5 h in aCSF at room temperature before use.

### 4.4. Electrophysiology

The hippocampal slices were transferred into a recording chamber (28–30 °C) and super fused with aCSF using a perfusion pump (perfusion rate, 2 mL/min). Two bipolar wires were twisted, and stimulus was applied on the pathway in parallel. A stimulus was applied to each position on the Schaffer collateral pathway (as the LTP input) [50], and an additional stimulus was applied to the subiculum region as a control. Metal microelectrodes (573210; science-products, A-M systems, WA, USA) were used in the slice recording of aCA1 pyramidal cells (Appendix A). Measurements were connected to the tungsten microelectrodes in SYS-DAM80 (World Precision Instruments, Sarasota, USA). For LTP induction, a stable baseline was recorded for 30 min; then, two trains of high-frequency stimulation (100 Hz, 1 s duration, with 30 s interval) were used for data acquisition. Five minutes after LTP induction, depotentiation was induced by theta-pulse stimulation (5 Hz, 1 min duration). The slopes of fEPSPs were analyzed using Win-LTP program version 2.32 (www.winltp.com).

### 4.5. Western Blot Analysis

Hippocampal slices were obtained in the same way as that used for the electrophysiology experiments. Samples were placed in protein lysis buffer (IBS-BR002; iNtRON Biotechnology, Gyeonggi-do, Korea) containing a protease inhibitor cocktail and homogenized with an ultrasonic grinder. The total concentration of the homogenized protein was measured using Bicinchoninic Acid (BCA) protein assay (23227; Thermo Scientific Pierce, CA, USA). Proteins were resolved according to their size on sodium dodecyl sulfate-polyacrylamide gels and transferred to polyvinylidene fluoride membranes (IPVH00010; Millipore, Bedford, MA, USA). The membranes were probed with primary antibodies against NRG1 (ab53104; Abcam, Cambridge, UK) and ErbB4 (MAI-861; Thermo Scientific Pierce) overnight, and then probed for 1 h with horseradish peroxidase-conjugated secondary antibodies. Immunoreactive bands were visualized using an enhanced chemiluminescence detection system (WBKLS0500; Millipore). Optical densities of the bands were quantified using ImageJ software version 1.53g (http://rsbweb.nig.gov/ij/).

### 4.6. Immunohistochemistry

Immunohistochemistry was performed as reported previously [30]. The mouse brains were fixed in 10% buffered formalin for 3 days. The brains were embedded in paraffin, and sliced tissues (a thickness of 4 µm) were stained with hematoxylin for histopathological evaluation. The slices were stained with a 1:1000 dilution of PV antibodies (PA1-933, Thermo Fisher, Rockford, IL, USA) and a 1:100 dilution of NeuN antibodies (Merck Millipore, Darmstadt, Germany) using an automated immunostainer (Bond-maX DC2002; Leica Biosystems, Bannockburn, IL, USA). For PV and NeuN antibodies, the heat-induced epitope retrieval was carried out using antigen retrieval buffer (Tris-ethylene diamine tetra acetic acid (EDTA) buffer, pH 9.0). Negative controls were treated accordingly in the absence of the primary antibody.

### 4.7. Object Recognition Test

In the novel object recognition test, six mice were exposed to several objects and their movements were recorded in a square open-field box (30 cm × 36 cm × 40 cm). The test was performed for a total of 5 days. In the habituation period, during the first 3 days, each mouse (both AD and WT groups) was placed in the empty box without objects for about 10 min. In the familiar session on the fourth day, the mice were exposed to two identical objects (blue low cylinder model), and the activity of the mice was analyzed. In the novel session on the fifth day, one familiar object was replaced with a new object (orange low cone model), and the behavior of the mice was recorded. This test was analyzed using the Nodule program, and the discrimination indices were calculated as follows:

PR (novel) = time (novel)/[time (novel) + time (familiar)] × 100

PR (familiar) = time (familiar)/[time (novel) + time (familiar)] × 100

### 4.8. Statistical Analysis

Data of the fEPSP slopes are shown as means ± standard errors of the mean and analyzed using the unpaired two-tailed Student’s *t*-test. Tukey’s multiple comparison test was used for the novel object recognition test. All analyses were performed using GraphPad Prism v.8 (GraphPad Software, San Diego, CA, USA).

## Figures and Tables

**Figure 1 ijms-22-00698-f001:**
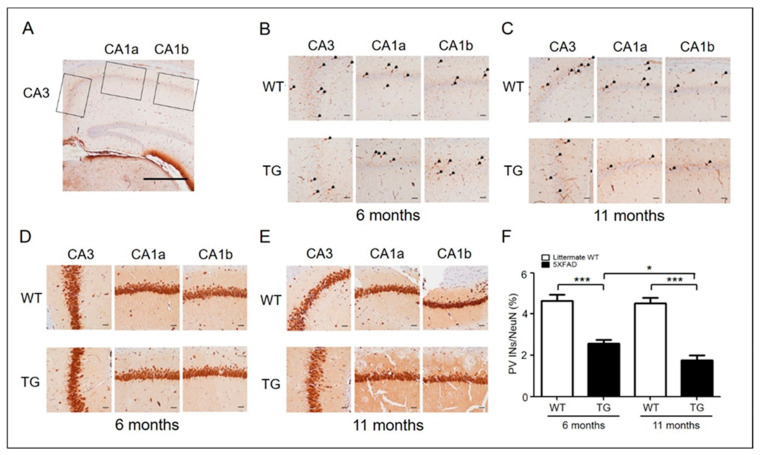
PV interneurons in AD mice (6-month-old adult and 11-month-old aged) and wild type. (**A**) Representative PV and NeuN immunohistochemical stains of the hippocampus CA3-CA1 region (original magnification, ×40) (scale bar = 500 μm). (**B**) Representative PV immunohistochemical stains of the hippocampus of 6-month-old (adult) WT (*n* = 6) and 5xFAD (*n* = 6) mice. PV-positive interneurons are shown in the hippocampus at a closer distance from CA1a-b (original magnification, × 200, pointed by arrowheads) and CA3 (original magnification, ×200) (scale bar = 200 μm). (**C**) Representative PV immunohistochemical stains of the hippocampus of 11-month-old (aged) WT (*n* = 6) and 5xFAD (*n* = 6) mice. (**D**) Representative NeuN immunohistochemical stains of the hippocampi of 6-month-old (adult) WT (*n* = 6) and 5xFAD (*n* = 6) mice. NeuN-positive cells are shown in the hippocampus at a closer distance from CA1a-b (original magnification, ×200, stained cells) and CA3 (original magnification, ×200) (scale bar = 200 μm). (**E**) Representative NeuN immunohistochemical stains of the hippocampus of 11-month-old (aged) WT (*n* = 6) and 5xFAD (*n* = 6) mice. (**F**) The ratio of PV-positive cells to NeuN-positive cells in adult and aged 5xFAD mice (*, *p* < 0.05; ***, *p* < 0.0001 compared with the indicated group using unpaired *t*-test). AD, Alzheimer’s disease; PV, parvalbumin; INs, interneurons; NeuN, neuronal nuclei; CA, cornu ammonis; TG, transgenic; WT, wild-type.

**Figure 2 ijms-22-00698-f002:**
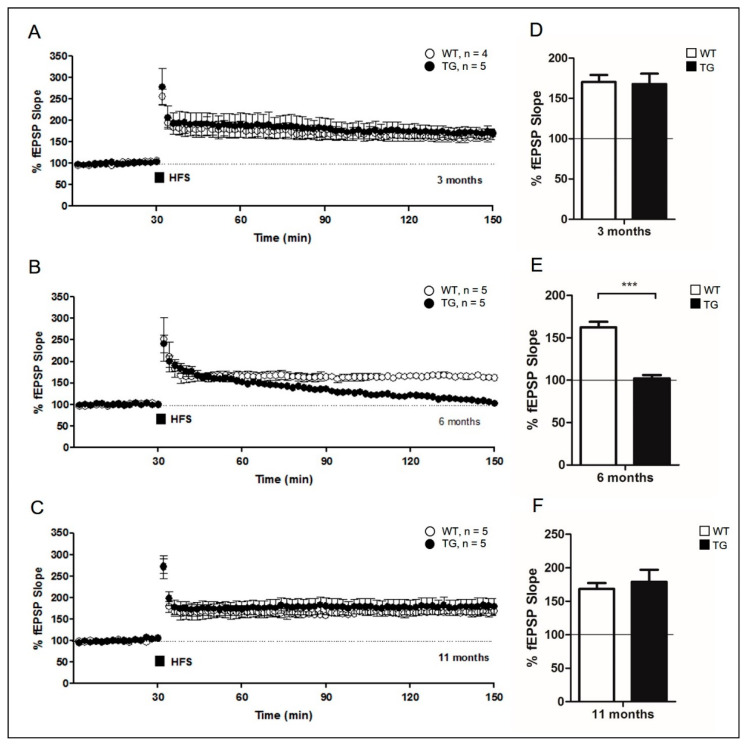
Evaluation of synaptic plasticity in aged AD mice. (**A**–**C**) Two trains of tetanus stimuli (100 Hz for 1 s, with a 30 s inter-tetanus interval) were applied to the Schaffer collateral pathway, which induced long-term potentiation in the hippocampal CA1 region. (**D**–**F**) Summary of these results. Data are displayed as the mean ± SEM (***, *p* < 0.0001 compared with the indicated group using unpaired *t*-test). fEPSP, field excitatory postsynaptic potential; WT, wild-type; TG, transgenic; HFS, high-frequency stimulation.

**Figure 3 ijms-22-00698-f003:**
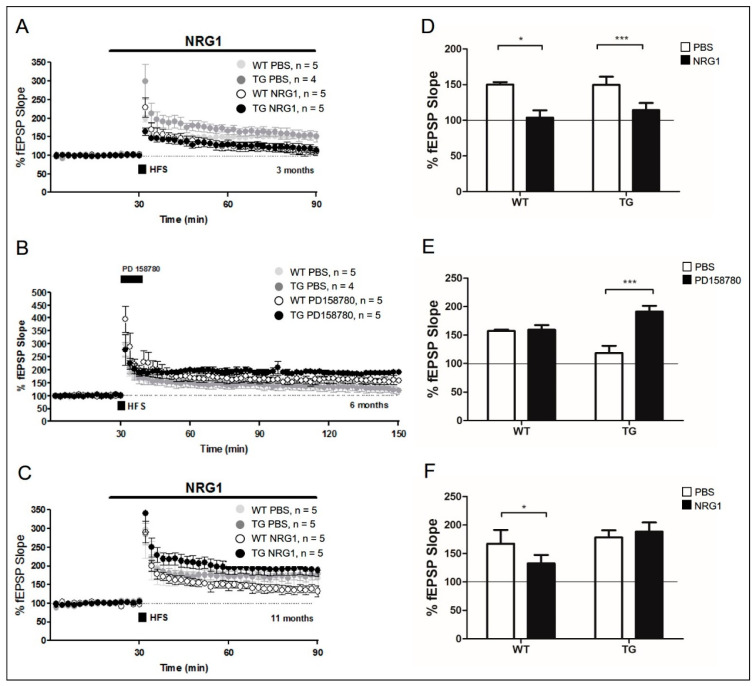
NRG1-expressed inhibition of LTP in aged 5xFAD mice. (**A**,**C**) Perfusion with 1 nM NRG1 peptide in 3- or 11-month-old WT and 5xFAD mice. (**B**) Perfusion with 10 μM PD158780 in 6-month-old WT and 5xFAD mice. (**D**–**F**) Summary of the results. Data are displayed as the mean ± SEM. *, *p* < 0.05; ***, *p* < 0.0001 compared with the indicated group using unpaired *t*-test. fEPSP, field excitatory postsynaptic potential; WT, wild-type; TG, transgenic.

**Figure 4 ijms-22-00698-f004:**
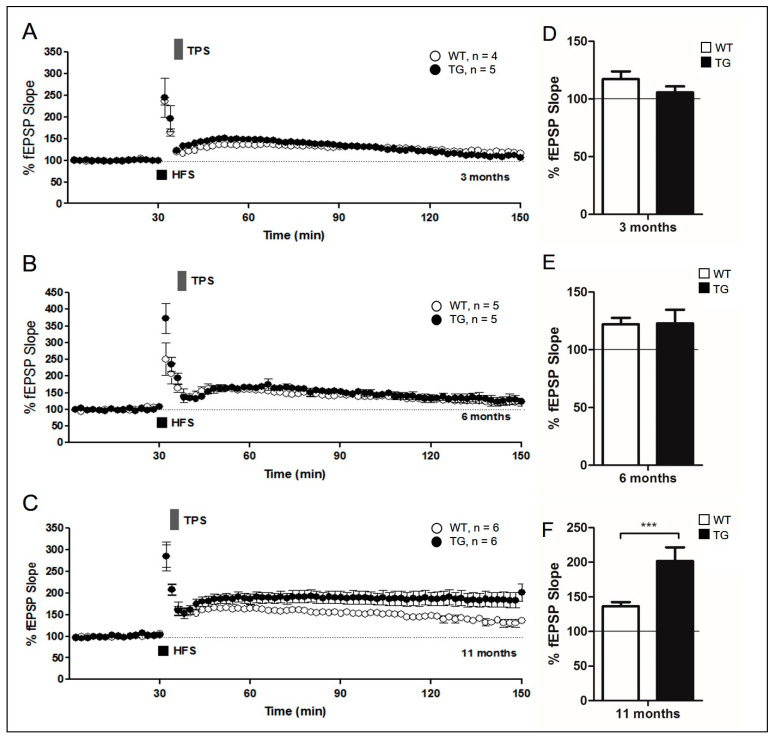
Evaluation of depotentiation -induced NRG1-ErbB signaling and theta-pulse stimulation (TPS)-induced impairment of depotentiation in aged 5xFAD mice. (**A**–**C**) Application of theta-pulse stimulation (TPS) (5 Hz for 1 min) 5 min after high-frequency stimulation (HFS) inverts LTP in WT and 5xFAD mice. (**D**–**F**) Summary of the results of the graph. Data are displayed as the mean ± SEM (***, *p* < 0.0001 compared with the indicated group using unpaired *t*-test). fEPSP, field excitatory postsynaptic potential; WT, wild-type; TG, transgenic.

**Figure 5 ijms-22-00698-f005:**
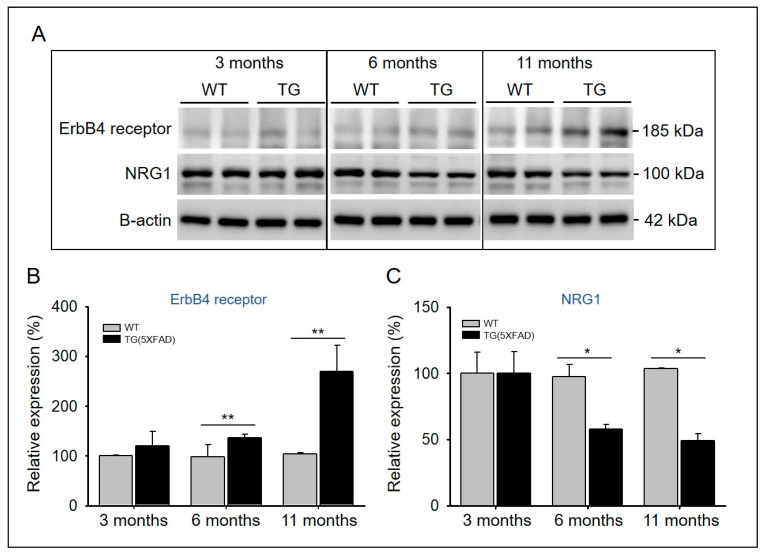
The expression levels of the NRG1 and ErbB4 receptor proteins. (**A**) The hippocampal expression levels of the NRG1 and ErbB4 receptors were performed with antibodies against NRG1, ErbB4 receptor, and β-actin (house keeping control). (**B**,**C**) The band intensities presented in panel A were measured by densitometry, normalized with those of β-actin, and expressed as relative fold-change in expressions. *, *p* < 0.05; **, *p* < 0.01 compared with the indicated group using unpaired *t*-test (*n* = 3–5).

**Figure 6 ijms-22-00698-f006:**
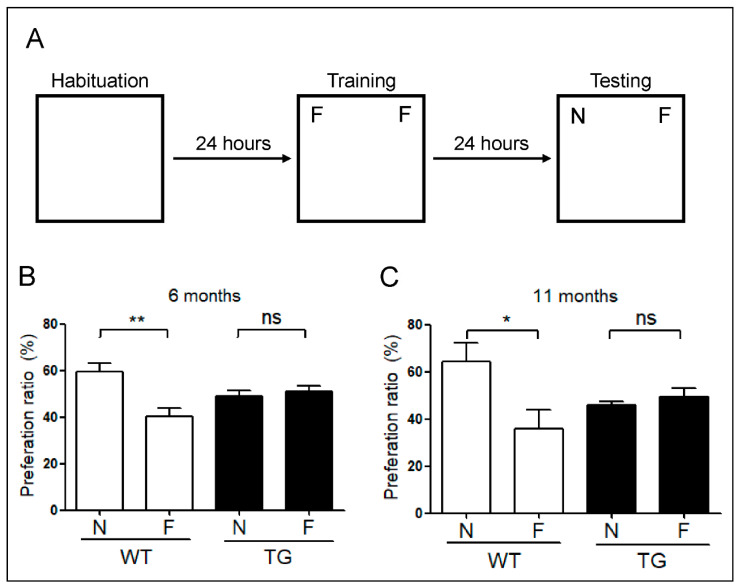
Object recognition test in 5xFAD mice. (**A**) Habituation training was performed for 3 days by exposing the animals to the experimental environment for 10 min a day in the absence of an object. A training session was conducted 24 h after the last habituation training. Mice were placed in the experimental setup in the presence of two identical objects and allowed to explore them for 10 min. After 24 h, the mice were returned to the experimental cage and observed while exploring the objects for 10 min. Two objects in the cage were familiar, and the third was a novel one. (**B**) Preference index of novel object recognition. In 6-month-old mice, the preference index levels were significantly different among WT, but not 5xFAD (TG) mice. (**C**) Preference index in the novel object recognition test. In 11-month-old mice, preference index levels were again significantly different among WT mice, whereas these levels did not differ among 5xFAD mice. Data are displayed as the mean ± SEM. *, *p* < 0.05; **, *p* < 0.01 relative to the indicated group using unpaired *t*-test (*n* = 6–10). F, familiar; N, novel; TG, transgenic; WT, wild-type; SEM, standard error of mean; ns, no significance.

**Figure 7 ijms-22-00698-f007:**
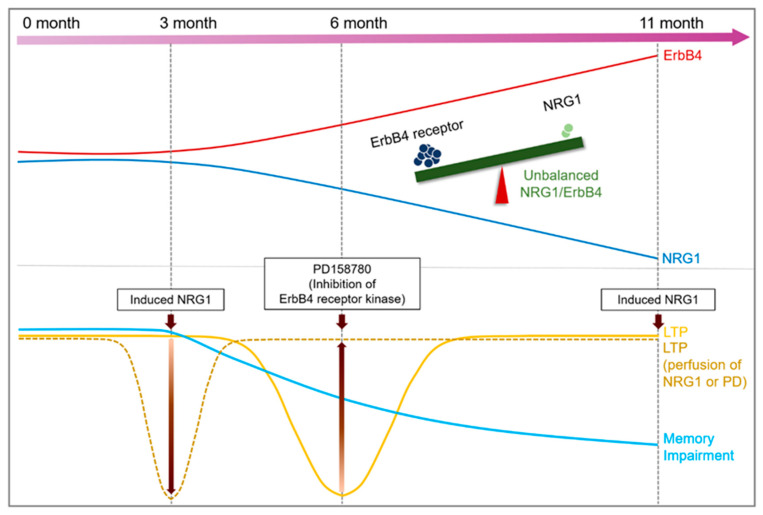
Schematic diagram of the hypothetical neuronal network of the time course in 5xFAD mice.

## Data Availability

Data are available on request from the corresponding author.

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
