# Peer review of "Inhibitory Neural Network’s Impairments at Hippocampal CA1 LTP in an Aged Transgenic Mouse Model of Alzheimer’s Disease"

_ijms, 2021, doi:10.3390/ijms22020698_

Round 1
Reviewer 1 Report
This paper explores the special role of NRG1-ErbB4 signaling in the development of AD by using 5xFAD mice. The findings are interesting, the experiments are well-organized and statistics is appropriate.
Major comments:
- The authors claimed that “We used mutant APP and PSEN1 expressing 5xFAD mice to investigate the pathology of Aβ induced neuronal dysfunction in AD at three different time points 3, 6, and 11 months” (Page 10, line 277-278). In figure 1, PV interneurons were only investigated at 6 months and 11 months. Do PV interneurons have different expressions in control and 5xFAD mice at 3 months? The authors said “5xFAD 39 mice, a model of AD with greater symptom severity than that of TG2576 mice” (Page 1, line 39). Could it possible that PV interneurons will decrease at 3 months?
- In figure 1, control results (WT) were presented first and experimental group (TG) were showed later. However, the sequence in Figure 2, Figure 3, Figure 4 was opposite. This is inappropriate. In Figure 3, PBS group (control) should also be at left side.
- “HFS-induced fEPSP enhancement returned to baseline at 1.5 hour 209 after the application of TPS in both young (3-month-old) 5xFAD and their WT littermates” (Page 7, line 208-209). However, the fEPSP is obviously higher than baseline at 120 min in Figure 4A.
- “The preference index did not indicate an impairment of object recognition memory in both 6- and 11-month-old 5xFAD mice” (Page 9, line 255-256). But the following results showed this memory was impaired.
- 5xFAD mice showed LTP impairment in 6 months and LTP reappearance at 12 months. Moreover, both 6 months and 12 months 5xFAD mice had memory impairment. Does that mean LTP is not the critical mechanism underlying memory impairment in AD model?
- Could the authors compare their findings in 5xFAD mice with their previous study which was done in TG2576 mice [29]? Does this study give any novel update?
Minor comments:
The authors should pay attention to check grammar mistakes. For example, “Moreover, HFS-induced LTP showeddepotentiation with TPS in aged littermate WT mice.” (Page 7, line 218)
Author Response
The authors wish to thank the reviewers for their helpful comments, and feel that these have contributed substantially to the substance of this manuscript.
Answers to Reviewers’ comments
# Reviewer 1
This paper explores the special role of NRG1-ErbB4 signaling in the development of AD by using 5xFAD mice. The findings are interesting, the experiments are well-organized and statistics is appropriate.
Major comments:
1) The authors claimed that “We used mutant APP and PSEN1 expressing 5xFAD mice to investigate the pathology of Aβ induced neuronal dysfunction in AD at three different time points 3, 6, and 11 months” (Page 10, line 277-278). In figure 1, PV interneurons were only investigated at 6 months and 11 months. Do PV interneurons have different expressions in control and 5xFAD mice at 3 months? The authors said “5xFAD 39 mice, a model of AD with greater symptom severity than that of TG2576 mice” (Page 1, line 39). Could it possible that PV interneurons will decrease at 3 months?
A: Previous studies (below references) reported that in the 5xFAD mice, “Neuron loss occurs in multiple brain regions, beginning at about 6 months in the areas with the most pronounced amyloidosis. Mice display a range of cognitive and motor deficits” (Oakley et al., 2006; Eimer and Vassar, 2013). Moreover, “Basal synaptic transmission and LTP in the hippocampal area CA1 begin to deteriorate between 4 and 6 months, but paired-pulse facilitation remains normal at least until 6 months” (Kimura and Ohno, 2009; Crouzin et al., 2013). Therefore, we assumed that PV interneurons were not decrease at 3 months.
-References-
(1) Oakley H, Cole SL, Logan S, Maus E, Shao P, Craft J, Guillozet-Bongaarts A, Ohno M, Disterhoft J, Van Eldik L, Berry R, Vassar R. Intraneuronal beta-amyloid aggregates, neurodegeneration, and neuron loss in transgenic mice with five familial Alzheimer's disease mutations: potential factors in amyloid plaque formation. J Neurosci. 2006 Oct 4;26(40):10129-40.
(2) Eimer WA, Vassar R. Neuron loss in the 5XFAD mouse model of Alzheimer's disease correlates with intraneuronal Aβ42 accumulation and Caspase-3 activation. Mol Neurodegener. 2013; 8:2.
(3) Kimura R, Ohno M. Impairments in remote memory stabilization precede hippocampal synaptic and cognitive failures in 5XFAD Alzheimer mouse model. Neurobiol Dis. 2009 Feb;33(2):229-35.
(4) Crouzin N, Baranger K, Cavalier M, Marchalant Y, Cohen-Solal C, Roman FS, Khrestchatisky M, Rivera S, Féron F, Vignes M. Area-Specific Alterations of Synaptic Plasticity in the 5XFAD Mouse Model of Alzheimer's Disease: Dissociation between Somatosensory Cortex and Hippocampus. PLoS One. 2013;8(9): e74667.
2) In figure 1, control results (WT) were presented first and experimental group (TG) were showed later. However, the sequence in Figure 2, Figure 3, Figure 4 was opposite. This is inappropriate. In Figure 3, PBS group (control) should also be at left side.
A: Following your suggestion, we have modified Figure 2~4 in the revised manuscript.
3) “HFS-induced fEPSP enhancement returned to baseline at 1.5 hour 209 after the application of TPS in both young (3-month-old) 5xFAD and their WT littermates” (Page 7, line 208-209). However, the fEPSP is obviously higher than baseline at 120 min in Figure 4A.
A: We have clarified the sentence as follows: “HFS-induced fEPSP enhancement decreased returning approximately to baseline after 2 hour the application of TPS in both young (3-month-old) 5xFAD and their WT littermates” (Page 7, line 209-211).
4) “The preference index did not indicate an impairment of object recognition memory in both 6- and 11-month-old 5xFAD mice” (Page 9, line 255-256). But the following results showed this memory was impaired.
A: We have changed the sentence as follows: “The preference index indicated an impairment of object recognition memory in both 6- and 11-month-old 5xFAD mice" (Page 9, line 256-257).
5) 5xFAD mice showed LTP impairment in 6 months and LTP reappearance at 12 months. Moreover, both 6 months and 12 months 5xFAD mice had memory impairment. Does that mean LTP is not the critical mechanism underlying memory impairment in AD model?
A: LTP is an important mechanism underlying general memory impairment, but in order to evaluate memory impairment specifically due to neural network damage in an AD model, the impairment must be assessed based on the results of several experiments, including behavioral tests. Therefore, when the behavioral experiment and LTP results are combined, we have concluded that the reappeared LTP in the 11-month-old TG mouse is not related to memory.
6) Could the authors compare their findings in 5xFAD mice with their previous study which was done in TG2576 mice [29]? Does this study give any novel update?
A: 5xFAD is an AD-related pathology model that overexpresses APP and PSEN1, and is characterized by more severe AD symptoms than TG2576 mice. Therefore, 1) PV loss was found at 11 months, earlier in 5XFAD mice (two mutations) than in TG2576 mice (one mutation) used in the previous study; 2) In the previous study used TG2576, we found evidence that NRG1-ErbB signaling affects the neuronal network, though the difference in expression levels of both proteins between WT and TG mice was not statistically significant (Huh et al., 2016). However, as a result of the western blot of 5xFAD mice used in this study, it was observed that the decrease of NRG1 protein and the increase of ErbB protein were statistically significant. Based on these results, we confirmed the impaired LTP in 5xFAD mice and its association with aberrant NRG1-ErbB signaling in the neuronal network to evaluate memory impairment.
-Reference-
Seonghoo Huh, Soo-Ji Baek, Kyung-Hwa Lee, Daniel J.Whitcomb, Jihoon Jo, Seong-Min Choi, Dong Hyun Kim, Man-Seok Park, Kun Ho Lee, Byeong C.Kim. The reemergence of long-term potentiation in aged Alzheimer;s disease mouse model. Scientific Reports. 2016; 6: 29152.
Minor comments:
1) The authors should pay attention to check grammar mistakes. For example, “Moreover, HFS-induced LTP showeddepotentiation with TPS in aged littermate WT mice.” (Page 7, line 218)
A: Thank you for your comment. We corrected our sentence like below following your comment “HFS-induced LTP showed LTP depotentiation with TPS in aged WT littermate mice.” (Page 7, line 219)
I hope our answers are satisfying.
Thank you very much.
Reviewer 2 Report
The authors should remove the sustantially contraindicatory statements between title and
abstract. In hippocampus we have intrinsinc inhibitory loops; for each subregion separately
and topographicaly arranged , which provide Lateral inhibition. Thus the damage of such loop
causes increase of action potentials frequency at Shaffer collaterals as well as at recurrent
collaterals of CA3 pyramidal cells. And increased frequency shoud cause the facilitation of LTP,
not impairment??? Please the authors to find out more in below metioned papers with the
exhaustive references, at last in methodological sections.
1. Computer Model of Synapse Loss During an Alzheimer’s Disease-Like Pathology in
Hippocampal Subregions DG, CA3 and CA1—The Way to Chaos and Information Transfer
Dariusz Świetlik, Jacek Białowąs, Janusz Moryś and Aida Kusiak
Entropy 2019, 21(4), 408; https://doi.org/10.3390/e21040408
2. Computer Modeling of Alzheimer’s Disease—Simulations of Synaptic Plasticity
and Memory in the CA3-CA1 Hippocampal Formation Microcircuit
Dariusz Świetlik, Jacek Białowąs, Janusz Moryś, Ilona Klejbor and Aida Kusiak
Molecules 2019, 24(10), 1909; https://doi.org/10.3390/molecules24101909
3. Effects of Inducing Gamma Oscillations in Hippocampal Subregions DG, CA3,
and CA1 on the Potential Alleviation of Alzheimer’s Disease-Related Pathology:
Computer Modeling and Simulations
Dariusz Świetlik, Jacek Białowąs, Janusz Moryś, and Aida Kusiak
Entropy 2019, 21(6), 587; https://doi.org/10.3390/e21060587
4. Schaffer Collateral Inputs to CA1 Excitatory and Inhibitory
Neurons Follow Different Connectivity Rules
Osung Kwon,1* XLinqing Feng,1* Shaul Druckmann,2 and XJinhyun Kim1,3
5140 • The Journal of Neuroscience, May 30, 2018 • 38(22):5140 –5152
This excellent paper (4) must be in references!!!
5. J. Physiol. (1973), 232, pp. 331-356 331
LONG-LASTING POTENTIATION OF SYNAPTIC TRANSMISSION IN THE DENTATE AREA
OF THE ANAESTHETIZED RABBIT FOLLOWING STIMULATION OF THE PERFORANT PATH
By T. V. P. BLISS AND T. L0MO
Authors should give the detailed positions of stimulating and recording electrodes, as
of Fig. 1 from the above paper (5.)
Other remarks :
1) The Authors give in Introduction the proper references (1 to 18), but they avoid any
discussion about neurofibrillary tangles patology, which is essential in first two stages
of AD patology from six after Braak classification, it requires completion!
2) About Neuregulin and LTP:
The LTP is induced by Calcium ions flow through NMDA channels, but expressed by
enhanced conductivity of AMPA channels. Thus the neuregulin experiments affect
the expression of LTP but not the induction ; it must be clearly pointed out.
Still remain unsolved questions, does previously not potentiated synapsę undergo
LTD? What about forgetting mechanisms, which are necessary for proper learnig
of new cues without catastrophic forgetting previously learned ones?
3) The title statement about crucial role of inhibitory networks for impairment of
learning and memory seems to be not justificated by results. Neuregulin receptors
expression has not direct influence for inhibitory loops, and parvalbumin interneurons
loss (in all subregions : DG, CA3, CA1) occurs due to profound patological changes in
late stages of AD in humans as well as in adult transgenic mice.
4) Thus the vague suggestion for title : Coincidence of Parvalbumin interneurons loss
and neuregulin dependent impairment of LTP expresion in transgenic mice Alzheimer
patology model ?
Author Response
The authors wish to thank the reviewers for their helpful comments, and feel that these have contributed substantially to the substance of this manuscript.
Answers to Reviewers’ comments
# Reviewer 2
1) The authors should remove the substantially contra-indicatory statements between title and abstract. In hippocampus we have intrinsic inhibitory loops; for each sub-region separately and topographically arranged, which provide Lateral inhibition. Thus the damage of such loop causes increase of action potentials frequency at Shaffer collaterals as well as at recurrent collaterals of CA3 pyramidal cells. And increased frequency should cause the facilitation of LTP, not impairment???
A: The increased frequency is an impaired form in Alzheimer's pathology (Świetlik, Białowąs et al. 2019). The action potential is important for LTP depolarization; however, it is not required under all LTP conditions (Lisman and Spruston 2005). Therefore, the increased frequency is not essential to trigger LTP. In addition, the mechanism for the role of increased frequency in LTP triggering requires further study.
- References-
Świetlik, D., J. Białowąs, J. Moryś, I. Klejbor and A. Kusiak (2019). "Effects of Inducing Gamma Oscillations in Hippocampal Subregions DG, CA3, and CA1 on the Potential Alleviation of Alzheimer's Disease-Related Pathology: Computer Modeling and Simulations." Entropy (Basel) 21(6).
Lisman, J. and N. Spruston (2005). "Postsynaptic depolarization requirements for LTP and LTD: a critique of spike timing-dependent plasticity." Nat Neurosci 8(7): 839-841.
2) Please the authors to find out more in below mentioned papers with the exhaustive references, at last in methodological sections. This excellent paper (4) must be in references!!!
A: According to your suggestion, we have cited the paper in Page 12, Line 365.
Świetlik, D., J. Białowąs, J. Moryś and A. Kusiak (2019). "Computer Model of Synapse Loss During an Alzheimer's Disease-Like Pathology in Hippocampal Subregions DG, CA3 and CA1-The Way to Chaos and Information Transfer." Entropy (Basel, Switzerland) 21(4): 408.
3) Authors should give the detailed positions of stimulating and recording electrodes, as of Fig. 1 from the above paper. “Long-lasting potentiation of synaptic transmission in the dentate area of the anaesthetized rabbit followings stimulation of the performant path” by T.V.P. Bliss and T. Lomo
A: According for your comment, we provided a new figure to show the precise position of the stimulating and recording electrodes, and we added Supplementary figure 2.
Please see the attached PDF file
Supplementary Figure 2. Schematic image of stimulating and recording electrodes. The field excitatory postsynaptic potentials (fEPSPs) was recorded with extracellular electrodes using mice hippocampal slices. The stimulating electrodes were placed in the Schaffer collateral pathway (Stimulation 1) and in the subiculum area (Stimulation 2). The recording electrode was placed in the CA1 pyramidal cells, and recorded the slope of fEPSPs.
Other remarks :
1) The Authors give in Introduction the proper references (1 to 18), but they avoid any discussion about neurofibrillary tangles patology, which is essential in first two stages of AD pathology from six after Braak classification, it requires completion!
A: According to your suggestion, we discuss the neurofibrillary tangles pathology in Page 2, line 55-57.
2) About Neuregulin and LTP: The LTP is induced by calcium ions flow through NMDA channels, but expressed by enhanced conductivity of AMPA channels. Thus, the neuregulin experiments affect the expression of LTP but not the induction; it must be clearly pointed out.
A: According to your comment, we modified the sentences as follows:
- A: (Page 2, line 82-84) ''Moreover, the treatment of hippocampal slices with NRG1 rapidly inhibits LTP induction in Schaffer collateral-CA1 synapses'' to ''Moreover, the treatment of hippocampal slices with NRG1 rapidly inhibits LTP expression in Schaffer collateral-CA1 synapses''
- (Page 5, line 159-160) "we perfused hippocampal slices with recombinant NRG1 peptide and observed the expression of hippocampal LTP by activating ErbB4 receptor kinase in the PV interneurons "
- (Page 5, line 168-169) "Perfusion with NRG1 in both 3-month-old 5xFAD and their WT littermates expressed a reduction of LTP and showed induction of NRG1-ErbB4 signaling."
- (Page 5, line 171) "We confirmed that NRG1 perfusion inhibited LTP expression in aged WT."
- (Page 5, line 175-176) "Our results indicate that the NRG1-expressed was not LTP-expressed fEPSP in aged 5xFAD,"
- (Page 6, line 186-187) Figure 3. "Neuregulin 1 (NRG1)-expressed inhibition of long-term potentiation."
- (Page 11, line 292-293) "Perfusion of NRG1 to the hippocampal slices prevents LTP expression at the CA1 synapses (Fig. 2A), "
3) Still remain unsolved questions, does previously not potentiated synapse, undergo LTD? What about forgetting mechanisms, which are necessary for proper leaning of new cues without catastrophic forgetting previously learned ones?
A: According to your comment, we modified the sentences in discussion section. (Page 11, line 302-305) "Unlike long term depression (LTD), depotentiation is the response reduction affecting the responses previously increased by LTP. In addition, depotentiation mechanisms are forgetting mechanisms, necessary for proper leaning of new cues without catastrophically forgetting previously learned ones (Sachser, Santana et al. 2016). " We have added reference, No 46.
4) The title statement about crucial role of inhibitory networks for impairment of learning and memory seems to be not justified by results. Neuregulin receptors expression has not direct influence for inhibitory loops, and parvalbumin interneurons loss (in all subregions: DG, CA3, CA1) occurs due to profound pathological changes in late stage of AD in human as well as in adult transgenic mice. Thus the vague suggestion for title: Coincidence of Parvalbumin interneurons loss and neuregulin dependent impairment of LTP expression in transgenic mice Alzheimer pathology model?
A: Thank you for your comment. After considering your suggestion, my co-authors and I came up with the following title: The role of NRG1-ErbB signaling in aberrant hippocampal synaptic functions in aged transgenic 5xFAD mice. However, we are concerned that this changed title and your suggested title are too narrow to cover our study. Therefore, we believe it would be better to keep the original title.

Round 2
Reviewer 1 Report
The authors answered my concerns.
Reviewer 2 Report
A: Thank you for your comment. After considering your suggestion, my co-authors and I came up with the following title: The role of NRG1-ErbB signaling in aberrant hippocampal synaptic functions in aged transgenic 5xFAD mice. However, we are concerned that this changed title and your suggested title are too narrow to cover our study. Therefore, we believe it would be better to keep the original title.
Dear Authors,
It is really not possible to cover with short title all aspects of broader work. But the art of work of inhibitory interneurons in all hippacampal region is very complicated, I suggest to read for improvement of further work some details about disinhibitory influence of theta rythm from medial septum neurons in a set of works of Vasilis Cutsuridis and Michael Hasselmo. For quick information I propose :
1) Septal networks: relevance to theta rhythm, epilepsy
and Alzheimer’s disease by Luis V. Colom
Journal of Neurochemistry, 2006, 96, 609–623
2)
Co-transmission of acetylcholine and GABA regulates hippocampal states
Virág T. Takács1, Csaba Cserép1,4, Dániel Schlingloff1,2, Balázs Pósfai1, András Szőnyi1,2, Katalin E. Sos1,2,
Zsuzsanna Környei3, Ádám Dénes3, Attila I. Gulyás1, Tamás F. Freund1 & Gábor Nyiri
DOI: 10.1038/s41467-018-05136-1 (Nature Communications)
3) Last important remark : The supplementary figure shoul be in main text, with some graphical improvements.
For me it is clear, but for readers with some litle knowledge of anatomy it seems to be better to show (may be schematically) paramidal cells with IE triangles and the course of Shaffer collaterals.